# Active Flow Matching

Yashvir S. Grewal[1,2], Daniel M. Steinberg[2], Edwin V. Bonilla[2], and
Thang D. Bui[1]

[1] Australian National University, Australia[1]
[2] Data61, CSIRO, Australia[2]

**Abstract.** Discrete diffusion and flow matching excel at capturing epistatic
structure in protein fitness landscapes through parallel, iterative refine-
ment. However, their implicit nature—sampling via learned dynamics
without tractable densities—prevents direct use with principled varia-
tional frameworks like VSD and CbAS for budget-constrained design. We
introduce *Active Flow Matching (AFM)*, which reformulates variational
objectives to operate on conditional endpoint distributions along the
flow rather than requiring $\log q_\phi(x)$. This enables gradient-based steer-
ing of flow models toward high-fitness regions while preserving the rigor
of VSD and CbAS. We derive forward-KL and reverse-KL variants us-
ing self-normalised importance sampling. Across four protein design tasks
forward-KL AFM consistently achieves lower regret and higher optimiza-
tion performance than VSD and diffusion-based LaMBO-2, demonstrat-
ing effective exploration-exploitation under tight experimental budgets.

## 1 Introduction

Autoregressive (AR) decoders are commonly used across domains for discrete
generation tasks, but their left-to-right factorisation cannot revise early tokens.
This is a fundamental mismatch for *epistatic* systems where changing position
$i$ alters the effect of changing $j$. Protein fitness landscapes exhibit such cou-
pling where distant residues interact through 3D folding and binding [Starr
and Thornton, 2016, Phillips, 2008]. The *fitness square* formalizes this: indepen-
dence requires $F_{11} = F_{10} + F_{01} - F_{00}$, but evolution frequently yields epistasis
$\varepsilon = F_{11} - F_{10} - F_{01} + F_{00} \neq 0$. Capturing $\varepsilon$ demands joint updates across sites.

Non-autoregressive iterative refinement models such as discrete diffusion and
flow matching, generate all positions in parallel, enabling global coupling [Austin
et al., 2021, Gat et al., 2024]. These models match or exceed AR/masked base-
lines across protein and RNA design (EvoDiff, DiMA, RFdiffusion, Chroma,
RNAdiffusion, DNA-Diffusion), and can also enable structure-conditioned gener-
ation (RFdiffusion, FoldFlow, motif-scaffolding) [Alamdari et al., 2023, Meshchani-
nov et al., 2024, Watson et al., 2023, Ingraham et al., 2023, Huang et al., 2024,
DaSilva et al., 2024, Bose et al., 2024, Trippe et al., 2022].

Translating these generative capabilities into practical discoveries requires
navigating finite experimental budgets. Discovery loops face combinatorial search
($20^{20} \approx 10^{26}$ for 20-residue peptides) and expensive experiments ($\sim$ \$500–2000

per assay) [Biophysics and Core, 2024, Core, 2024, for Macromolecular Interactions, 2024, GenScript, 2025]. We adopt *active generation* view to solve this problem, where we learn $q(x \mid y > \tau)$, the conditional distribution of high-fitness designs under fixed budgets [Steinberg et al., 2025a]. Practical requirements include (i) diverse batches for parallel screening [Jain et al., 2023, Steinberg et al., 2025b], (ii) multi-objective flexibility [Stanton et al., 2022, Jain et al., 2023], and (iii) interpretable structure discovery via co-occurrence patterns in the batch [Marks et al., 2011, Hopf et al., 2014].

Two principled approaches cast active generation as variational inference over rare events. **VSD (reverse KL)** minimizes $\mathrm{KL}(q_\phi(x) \,\|\, p(x \mid y \geq \tau, D_t))$, yielding an ELBO with prior, likelihood, and entropy terms; discrete sequences require score-function estimators, demanding access to $\nabla_\phi \log q_\phi(x)$ [Steinberg et al., 2025a]. **CbAS (forward KL)** minimizes $\mathrm{KL}(p(x \mid y \geq \tau) \,\|\, q_\phi(x))$, yielding weighted MLE $\mathbb{E}_{p(x)}[w(x) \log q_\phi(x)]$ where $w(x) \propto \Pr(y \geq \tau \mid x)$ [Brookes et al., 2019]. Both support informative priors and batch-sequential updates, but both require tractable $q_\phi(x)$.

State-of-the-art discrete diffusion and flow-matching models are *implicit* generators: they optimise score/denoising or flow-regression objectives rather than a tractable likelihood, and thus do not yield normalised densities over discrete sequences. Consequently, evaluating or differentiating $\log q_\phi(x)$ is generally intractable. These models sample via learned dynamics but lack a usable mass function $q_\phi(x)$: for discrete diffusion, exact $\log q_\phi(x)$ requires summing over exponentially many corruption paths [Austin et al., 2021]; for discrete flow matching, current formulations provide no simple closed-form mass function [Lipman et al., 2022, Gat et al., 2024]. Objectives that require $\log q_\phi(x)$ or its score $\nabla_\phi \log q_\phi(x)$ are therefore incompatible with these generative models.

*Active Flow Matching (AFM).* We resolve this by reformulating variational objectives to operate on conditional endpoint distributions along the flow rather than on $q_\phi(x)$ itself. Active Flow Matching preserves the principled foundations of VSD and CBAS while leveraging implicit generators for principled, budget-efficient design.

## 2   Active Flow Matching (AFM)

*Setup.* Let $\mathcal{X} = \Sigma^L$ denote the sequence space. We train a discrete-state flow that induces, for each $t \in [0, 1]$, the conditional endpoint distribution $q_t^\theta(\mathbf{x}_1 \mid \mathbf{x}_t)$. The flow starts from uniform $u(\mathbf{x}) = |\Sigma|^{-L}$ at $t = 0$. A class probability estimator provides scores $p(y{=}1 \mid \mathbf{x}, \mathcal{D})$ for desirable sequences, where $y$ denotes the property label.

*Forward-KL AFM* If we could sample from $p(\mathbf{x}_1|y)$, we would learn the flow by simply minimising

$$\mathcal{L}_{\mathrm{gVFM}}(\theta) = \mathbb{E}_{t,\mathbf{x}_t|y} \left[ \mathrm{KL} \left[ p_t(\mathbf{x}_1|\mathbf{x}_t, y) \| q_t^\theta(\mathbf{x}_1|\mathbf{x}_t) \right] \right] = -\mathbb{E}_{t,\mathbf{x}_1|y,\mathbf{x}_t} \left[ \log q_t^\theta(\mathbf{x}_1|\mathbf{x}_t) \right] + \mathrm{const.} \tag{1}$$

Since sampling from $p(\mathbf{x}_1|y)$ is intractable, we use self-normalized importance sampling (SNIS) with a proposal distribution $q(\mathbf{x}_1)$:

$$\mathcal{L}_{\mathrm{gVFM}}(\theta) = -\mathbb{E}_{t,\mathbf{x}_1 \sim q(\mathbf{x}_1),\mathbf{x}_t} \left[ \frac{p(\mathbf{x}_1|y)}{q(\mathbf{x}_1)} \log q_t^\theta(\mathbf{x}_1|\mathbf{x}_t) \right] \tag{2}$$

$$\approx -\mathbb{E}_{t,\mathbf{x}_t} \left[ \frac{\sum_{k=1}^K w_k \log q_t^\theta(\mathbf{x}_{1,k}|\mathbf{x}_t)}{\sum_{k=1}^K w_k} \right], \tag{3}$$

where $\{\mathbf{x}_{1,k}\}_{k=1}^K \sim q(\mathbf{x}_1)$, $w_k = \frac{p(\mathbf{x}_{1,k},y)}{q(\mathbf{x}_{1,k})}$, $t \sim \mathrm{Unif}[0,1]$, and $\mathbf{x}_t$ is sampled from the model's CTMC.

*Reverse-KL AFM*   At each round, we steer the base flow (from the previous round) toward high-property regions by minimizing

$$\mathcal{L}_{\mathrm{srVFM}}(\phi) = \mathbb{E}_{t,\mathbf{x}_t} \left[ \mathrm{KL} \left[ q_t^\phi(\mathbf{x}_1|\mathbf{x}_t) \| p_t(\mathbf{x}_1|\mathbf{x}_t) \right] \right], \tag{4}$$

where $p_t(\mathbf{x}_1|\mathbf{x}_t) \propto q_t^\theta(\mathbf{x}_1|\mathbf{x}_t)p(y|\mathbf{x}_1,\mathcal{D})$ and $\theta$ denotes the base flow parameters. Using SNIS with proposal $q(\mathbf{x}_1)$ yields:

$$\mathcal{L}_{\mathrm{srVFM}}(\phi) \approx \mathbb{E}_{t,\tilde{\mathbf{x}}_1 \sim q(\tilde{\mathbf{x}}_1),\mathbf{x}_t} \left[ \frac{p(\tilde{\mathbf{x}}_1|y)}{q(\tilde{\mathbf{x}}_1)} \mathbb{E}_{q_t^\phi(\mathbf{x}_1|\mathbf{x}_t)} \left[ \log q_t^\phi(\mathbf{x}_1|\mathbf{x}_t) - \log q_t^\theta(\mathbf{x}_1|\mathbf{x}_t) - \log p(y|\mathbf{x}_1,\mathcal{D}) \right] \right] \tag{5}$$

$$\approx \sum_{k=1}^K \tilde{w}_k \, \mathbb{E}_{q_t^\phi(\mathbf{x}_1|\mathbf{x}_{t,k})} \left[ \log q_t^\phi(\mathbf{x}_1|\mathbf{x}_{t,k}) - \log q_t^\theta(\mathbf{x}_1|\mathbf{x}_{t,k}) - \log p(y|\mathbf{x}_1,\mathcal{D}) \right], \tag{6}$$

where $\tilde{w}_k = w_k / \sum_{k'=1}^K w_{k'}$, $w_k = \frac{p(\mathbf{x}_{1,k},y)}{q(\mathbf{x}_{1,k})}$, $t \sim \mathcal{U}(0,1)$, $\mathbf{x}_{1,k} \sim q(\mathbf{x}_1)$.

Our choice of proposal distribution (a mixture of unlabelled data, flow endpoints, and a replay buffer) provides good coverage as demonstrated in the strong experiment results.

*Symmetric-KL baseline.* For completeness, we also report a symmetric-KL variant that adds the forward- and reverse-KL objectives above, implemented with the same SNIS endpoint sampling scheme.

## 3   Experiments

We evaluate on four protein design tasks: Ehrlich synthetic objectives (lengths 32, 64) [Stanton et al., 2024], FoldX stability [Guerois et al., 2002, Schymkowitz et al., 2005, Delgado et al., 2019], SASA optimization [Lee and Richards, 1971, Shrake and Rupley, 1973, pol]. We compare Forward-KL AFM against VSD and diffusion-based LaMBO-2. For tasks with known optima (Ehrlich), we report simple regret $r_t = f(x^\star) - \max_{1 \le s \le t} f(x_s)$. For unknown optima (FoldX, SASA),

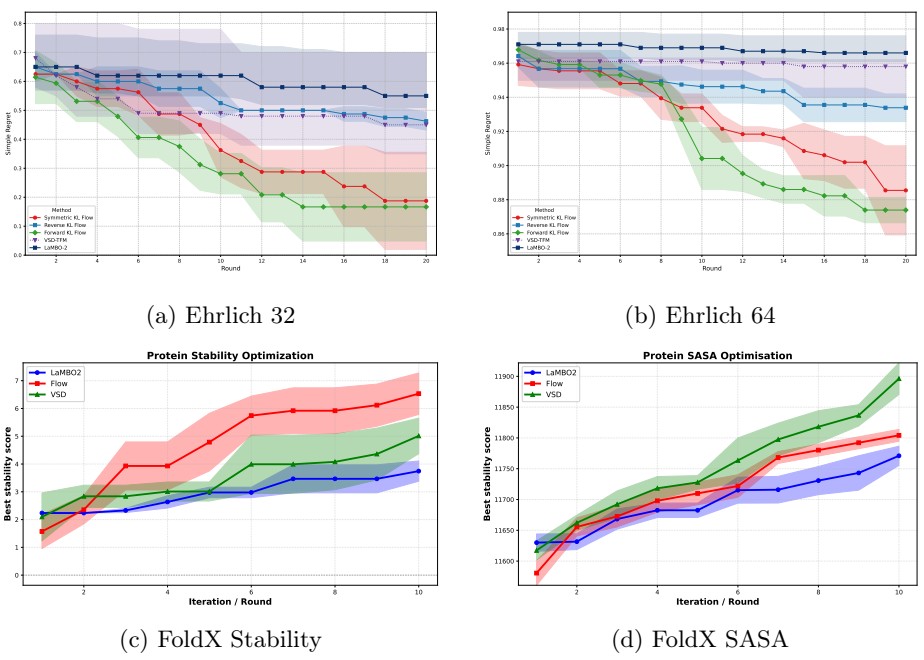

(a) Ehrlich 32

(b) Ehrlich 64

(c) FoldX Stability

(d) FoldX SASA

Fig. 1: Performance comparison across four protein design tasks. Forward-KL AFM achieves superior optimization compared to VSD and Lambo2 baselines.

we report highest value uptil each round $I_t = \max_{1 \leq s \leq t} f(x_s)$. We use a batch size of 128 in ehrlich sequences and batch size of 10 in FoldX experiments

Forward-KL AFM achieves lowest regret (Ehrlich) and highest scores in FoldX stability. (Figure 1;). Reverse-KL's performs relatively poorly. Symmetric-KL performs competitively but trails Forward-KL on Ehrlich-32/64, indicating Forward-KL's mass-covering better balances exploration-exploitation in these sequence spaces.

## 4    Conclusion

We introduced Active Flow Matching (AFM), which enables principled variational optimization with implicit discrete generators by reformulating objectives on conditional endpoint distributions. This resolves the incompatibility between state-of-the-art flow models and likelihood-based frameworks like VSD and CbAS, allowing gradient-based steering without tractable $q_\phi(x)$. Across protein design tasks, forward-KL AFM consistently outperforms existing methods under tight experimental budgets, demonstrating effective exploration-exploitation. Our framework opens the door to leveraging powerful pretrained flow and diffusion models (e.g., EvoDiff, ESM-2) for budget-constrained discovery in proteins.

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
