# OpenReview forum: "Active Flow Matching"
_AJCAI/2025/Workshop/AIML-CEB — AIML-CEB 2025 Oral_

### Official Review · Reviewer_E2Ed · 2025-11-07

**Rating:** 9
**Confidence:** 4

**Review:**

This paper introduces Active Flow Matching (AFM), a novel framework that bridges the gap between implicit discrete generative models  and variational optimization frameworks (i.e., VSD, CbAS). The method builds on existing theory but extends it elegantly to a practical domain, protein and sequence design under constrained budgets.

The paper is technically rigorous, deriving both forward-KL and reverse-KL objectives clearly, and implements self-normalized importance sampling (SNIS) to approximate intractable distributions.

The experimental setup is credible: four protein design tasks are well-chosen benchmarks to demonstrate efficiency and exploration-exploitation balance.

Thank you for submitting this work to the AIML-CEB workshop. It will make a valuable addition to the workshop, stimulating discussion on integrating active learning, flow matching, and variational inference in generative biology.

**Suggestions for Improvement**

While overall strong, the following refinements could enhance the work:
1. **Ablation Analysis.** A comparison between forward- and reverse-KL behavior is shown, but more discussion on why forward-KL works better (i.e., mode coverage) would deepen the insight.
2. **Scalability and Generalization.** Discussion of how AFM scales to larger proteins or higher-dimensional latent spaces would help readers assess its applicability to real-world problems.
3. **Unclear Hyperparameter Sensitivity.** The performance may depend on design choices such as flow network architecture, batch size, or importance-sampling variance. An ablation study showing the impact of these factors would enhance reproducibility.

---

### Official Review · Reviewer_EzKB · 2025-11-08
**Active Flow Matching optimises protein fitness landscapes and outperforms baseline on tight experimental budgets**

**Rating:** 7
**Confidence:** 2

**Review:**

Protein sequence and 3D generation is a complex problem due to a large combinatorial search and costly experimental validation to guide and validate designs. Autoregression (AR) is not well adapted to generative tasks for proteins where AR’s linear prediction along a sequence won’t capture the potential dependency of one residue with residues that precede it. Discrete diffusion and flow matching generate all positions in parallel, therefore solve this issue. Additionally, these methods enabled structure-conditioned generation: generating a protein in presence of another molecule or using a fixed scaffold. However computational/experimental budgets are finite. The authors propose Active Flow Matching (AFM), which enables efficient exploration-exploitation balance under tight experimental budgets. They describe 3 variants and select the Forward-KL AFM for comparison with baselines. Four different generative task experiments: Erlich synthetic objectives 32 and 64, FoldX stability, SASA optimisation. For each experiment, the authors compare Forward-KL AFM (their method) against VSD and diffusion-based LaMBO-2.

Comments:
* The background, where this work positions itself and what it aims to solve was well laid out. It would have helped to provide a short sentence description of each of the 4 datasets used.
* Methods naming convention could be more consistent: in the text AFM is used to describe the author’s method, in the Fig1 Flow is used. Additionally, in the text, the authors mention VSD and LaMBO-2, and in other parts of the text VSD and CbAS. Are LaMBO-2 and CbAS the same?
* On the same topic as naming convention: colour convention is arbitrary in Fig1 across a,b,c,d. My understanding is that AFM forward-KL is performing best in a,b and that’s the method used under “Flow” in c,d. These should all have the same name + colour as they’re within the same figure.
* The focus of the Fig1 is on performance and the text describes “Across protein design tasks, forward-KL AFM consistently outperforms existing methods under tight experimental budgets”. However, this isn’t the case for Fig1c SASA optimisation, with VSD outperforming AFM after 10 rounds. Not a big deal but the word “consistently” can’t be used as a result.
* My understanding is that the suggestion that AFM outperforms methods under tight experimental budget is illustrated by the fact that only a “small” number of “rounds” are explored. If this is indeed correct, I’d suggest to emphasize this fact and highlight that AFM very quickly optimises proteins after very few rounds. Again if my assumption was correct, it would be interesting to let these methods generate proteins over a much larger number of rounds: does AFM plateau and eventually gets caught up by other methods?

---

### Official Review · Reviewer_Fyz2 · 2025-11-11
**Combines two recent advances in ML, well motivated for protein design**

**Rating:** 8
**Confidence:** 5

**Review:**

## Summary
The manuscript combines two ideas. First the recently proposed active generation framework, and second
the popular flow matching approach. A key advance is to replace the proposal density in active generation
with conditional endpoint distributions along the flow. This enables variational inference approaches
to be applied for generation of discrete sequences. Empirical experiments on protein design show the
method's promise.

## Evaluation
This well written manuscript advances some recently proposed ideas in machine learning, and is well motivated
by the problem of protein design.

Suggestions for improvement:
- The legends in Figure 1 are not clear, as to which method is AFM.
Also confusing why (a) and (b) have 5 methods while (c) and (d) only have 3.
- I would have liked a clearer explaination of what is conditional endpoint distributions,
and how they are plugged in to replace q_\phi(x). It would be useful to understand what is different in AFM
in comparison to VSD.

---

### Decision · Program_Chairs · 2025-11-12

Accept (Oral)